# What are the features of high-performing quality improvement collaboratives? A qualitative case study of a state-wide collaboratives programme

James G McGowan [1], Graham P Martin [1], Greta L Krapohl [2], Darrell A Campbell [2], Michael J Englesbe [2], Justin B Dimick [2], Mary Dixon-Woods [1]

¹The Healthcare Improvement Studies Institute (THIS Institute), Department of Public Health and Primary Care, University of Cambridge, Cambridge, UK
²Department of Surgery, University of Michigan, Ann Arbor, Michigan, USA

**Correspondence to**
Dr James G McGowan;
james.mcgowan@thisinstitute.cam.ac.uk

## ABSTRACT

**Objectives** Despite their widespread use, the evidence base for the effectiveness of quality improvement collaboratives remains mixed. Lack of clarity about 'what good looks like' in collaboratives remains a persistent problem. We aimed to identify the distinctive features of a state-wide collaboratives programme that has demonstrated sustained improvements in quality of care in a range of clinical specialties over a long period.

**Design** Qualitative case study involving interviews with purposively sampled participants, observations and analysis of documents.

**Setting** The Michigan Collaborative Quality Initiatives programme.

**Participants** 38 participants, including clinicians and managers from 10 collaboratives, and staff from the University of Michigan and Blue Cross Blue Shield of Michigan.

**Results** We identified five features that characterised success in the collaboratives programme: learning from positive deviance; high-quality coordination; high-quality measurement and comparative performance feedback; careful use of motivational levers; and mobilising professional leadership and building community. Rigorous measurement, securing professional leadership and engagement, cultivating a collaborative culture, creating accountability for quality, and relieving participating sites of unnecessary burdens associated with programme participation were all important to high performance.

**Conclusions** Our findings offer valuable learning for optimising collaboration-based approaches to improvement in healthcare, with implications for the design, structure and resourcing of quality improvement collaboratives. These findings are likely to be useful to clinicians, managers, policy-makers and health system leaders engaged in multiorganisational approaches to improving quality and safety.

## INTRODUCTION

Often addressing areas of clinical practice with known variation in care processes or outcomes, or gaps between current and best practice, quality improvement collaboratives seek to bring together healthcare professionals from multiple organisations to address shared improvement goals.[1–4] Though their structure and methods can vary, collaboratives typically seek to provide a mechanism to accelerate the uptake of practice innovations to improve care across multiple clinical settings.[4] Key activities include the provision of performance feedback to participating clinicians and organisations, usually benchmarked against data from other sites,[5–7] and face-to-face meetings to share learning and identify interventions that might be used to support and scale improvement.[2] Building on early models such as the Northern New England Cardiovascular Disease Study Group and Vermont Oxford Network in the 1980s,[5 8 9] and more recent approaches such as the Institute for Healthcare Improvement's

## STRENGTHS AND LIMITATIONS OF THIS STUDY

⇒ Identifying the drivers of successful quality improvement collaboratives is a priority for healthcare policy-makers, clinicians and researchers.
⇒ A qualitative design was appropriate to our goal of characterising the features of a high-performing collaboratives programme.
⇒ Our sample of clinicians and managers was constructed to capture variability in perspectives among participants.
⇒ Practical considerations meant it was not possible to sample from all collaboratives in the programme; purposive sampling of participants meant that some selection bias may have occurred.
⇒ Our study did not include patient perspectives, which may have differed from those of the study participants.

Breakthrough Series design,[10] collaboration-based approaches have emerged as a prominent strategy for improving healthcare quality and safety across OECD (Organisation for Economic Co-operation and Development) countries, including in the USA.[3 11] Despite their widespread use across a variety of clinical specialties and organisational contexts, however, the evidence base for the effectiveness of improvement collaboratives is mixed.[1–3 5–7 12 13]

One challenge in improving the consistency with which collaboratives improve quality of care arises from ongoing uncertainty regarding the underlying drivers of their performance, including lack of clarity about 'what good looks like'.[11 14–17] An emerging literature[18–20] suggests that understanding what characterises high performance is likely to help in addressing this problem. The literature, including research on high-reliability organisations,[21 22] the 'Safety II' movement (which emphasises the need to investigate how things can and do go well)[23–26] and the body of work on the importance of recognising and understanding 'positive deviance',[27] is united by its emphasis on the value of producing learning by identifying the features of high performance and making them accessible and replicable. An example is the use of positive deviance approaches[27 28] to address unwarranted variation in processes of care, as in the well-known example of door-to-balloon studies for cardiac catheterisation.[20 29]

In this article, we report a qualitative case study of a regional (state-wide) collaboratives programme that has demonstrated high performance over a long period: the Michigan Collaborative Quality Initiatives (CQIs) programme.[30] Funded by Blue Cross Blue Shield of Michigan, the programme operates multiple quality improvement collaboratives across a range of medical and surgical specialties. Focusing on care processes associated with high volume, cost and variation in practice, the programme is an important example of a collaborative model that has achieved sustained improvements in quality and safety, outperforming secular trends and improvements made by other improvement programmes.[31–35] Between 2005 and 2009, for example, hospitals participating in the general surgery collaborative reduced risk-adjusted morbidity by 2.6%, outperforming results achieved by the American College of Surgeons' National Surgical Quality Improvement Program (NSQIP).[31] Similarly, hospitals participating in the bariatric collaborative reduced risk-adjusted mortality faster than hospitals outside Michigan that participated in NSQIP during the same period.[31] More recent improvements have been reported by the urological[32 33] and cardiothoracic surgery[34] collaboratives.

We aimed to identify distinctive features of this state-wide programme that might be implicated in success and thereby build understanding of what good looks like for quality improvement collaboratives.

## METHODS

We conducted a qualitative case study of the CQIs programme based on semistructured interviews with participating clinicians and managers, observation of a quality improvement meeting of the Michigan Surgical Quality Collaborative (the largest and longest-standing CQI), and analysis of programme documents.

### Sampling and recruitment

We purposively sampled clinicians and managers across a spectrum from more mature to less well-established collaboratives. We sought to capture variation across a range of participant characteristics, including state geography, socioeconomic status of patient population served, size of participating hospital, urban and rural settings, and academic and non-academic hospitals. As a very high proportion of collaboratives in the CQI programme are surgical in nature, the ability to sample clinical participants from non-surgical specialties was more limited.

We also sampled participants from the host organisation of the collaboratives' coordinating centres (University of Michigan) and from the programme funder (Blue Cross Blue Shield of Michigan). A letter of invitation and study information document were sent to potential interview participants identified by GLK, all of whom were offered an opportunity to discuss the study with JGM before consenting to be interviewed.

### Data collection

Interviews were conducted in person or remotely (by telephone) by one interviewer (JGM), who was not known to participants, at coordinating centre sites and participating hospitals. We obtained written informed consent from all participants. Interviews were digitally recorded using an encrypted voice recorder and were supported by a topic guide (online supplemental file 2) that was informed by literature review and discussion within the project team. Audio recordings were transcribed and transcripts were anonymised.

Additionally, participant observation was conducted by JGM during a Michigan Surgical Quality Collaborative (MSQC) meeting. JGM attended individual sessions, observed presentations and interactions, and talked informally with other attendees about the work of the collaborative. This was supported by contemporaneous digital recordings made by JGM, which were subsequently transcribed, anonymised and included in the analysis. Programme documents were collected during the MSQC meeting and treated as data.

### Analysis

Data management was supported by NVivo software (V.10). Data analysis was based on the constant comparative method.[36] Each interview transcript was read, coded and recoded multiple times to describe each relevant unit of meaning. Comparisons were made between data, codes and categories in order to identify distinct analytic concepts in the data and relationships between them. A

**Table 1** Descriptive characteristics of interview participants

| Characteristic | Number of participants |
|---|---|
| Professional role | |
| Surgeons and physicians | 15 |
| Nurses | 11 |
| Programme managers | 12 |
| Organisation | |
| Michigan Surgical Quality Collaborative | 15 |
| Michigan Bariatric Surgery Collaborative | 5 |
| Blue Cross Blue Shield of Michigan | 5 |
| Michigan Arthroplasty Registry Collaborative Quality Initiative | 2 |
| Michigan Urological Surgery Improvement Collaborative | 2 |
| Michigan Oncology Quality Consortium | 2 |
| University of Michigan Quality Department | 2 |
| Michigan Trauma Quality Improvement Programme | 1 |
| Michigan Radiation Oncology Quality Consortium | 1 |
| Michigan Anticoagulation Quality Improvement Initiative | 1 |
| Michigan Society of Thoracic and Cardiovascular Surgeons Quality Collaborative | 1 |
| Michigan Cardiovascular Consortium | 1 |
| Total | 38 |

thematic framework was developed that was iteratively refined, theorised and applied to the data to synthesise findings. Data analysis and interpretation and development of the thematic framework were led by JGM with support from GPM and MD-W.

### Patient and public involvement
None.

### RESULTS
We included 38 interview participants (clinicians and managers from 10 CQIs, University of Michigan and Blue Cross Blue Shield of Michigan; table 1), 12 hours of participant observation and 12 programme documents in our study.

Our analysis identified five distinctive features implicated in the success of the CQIs programme: learning from positive deviance; high-quality coordination; high-quality measurement and comparative performance feedback; careful use of motivational levers; and mobilising professional leadership and building community. Online supplemental table 1 summarises the features and data categories identified by our analysis with supporting data.

### Learning from positive deviance
The collaboratives did not seek to promote a particular quality improvement methodology. Instead, consistent with the principles of positive deviance,[28] they focused on sourcing effective interventions to address specific quality problems from within the community, and then making them easy to share and implement widely. Accordingly, participants described the collaborative 'theory of change' as identifying positive outliers on measures of quality (which may be organisations, whole clinical teams or individual care providers), learning from them, making accessible the knowledge and practices associated with their high performance, and then promoting standardisation of practices to improve quality across the collaborative as a whole.

> We do it more, positive deviancy stories, and that's pretty common, I think, to most of the collaboratives. Try and go for people who have done, who are doing something really well, who have really exemplary performance…on outcomes … We systematise some of the knowledge, and present it back. (Surgeon)

In seeking out the clinical and management practices associated with high-quality care, collaborative leaders aimed to demonstrate the value of mutual learning, role-modelled openness to learning from colleagues and sites across the state, and deliberately enfranchised participants from non-academic centres and those not traditionally perceived as high performing.

> There are folks out there who do a phenomenal hip, there are folks out there who do a phenomenal colon. They don't have to be in a university set-up… Then, let's go to these places where we see an outlier surgeon doing exceptionally well, and see what he's doing different. (Surgeon)

### High-quality coordination
The organisation and actions of coordinating centres were consistently identified as important to the success of the collaboratives. Comprising clinically led teams with expertise in quality improvement, these centres met the definition of coordination as 'integrating or linking together different parts of an organisation to accomplish a collective set of tasks.'[37] One of their important roles was that of resolving mundane challenges that might otherwise undermine participation in collaborative quality improvement, for example, by providing technical support to data abstractors and clinicians at hospital sites, addressing data collection problems, conducting analysis and interpretation of clinical registry data, and supporting evaluation of interventions. By absorbing much of the hard work involved in improvement, including administrative tasks, coordinating centres reduced the load on staff at local sites.

> When they [the coordinating centre] are asking the abstractors to collect the information out of the

patients' charts to put it into the MSQC [Michigan Surgical Quality Collaborative] workstation, they have done tremendous hard work in making sure that they do not present it to the abstractors until it is truly ready to go in a useable form and that if we have questions, they have already done the hard work of answering those questions. (Nurse)

Participants proposed that by centralising responsibility for labour-intensive or technically challenging tasks such as data analysis, coordinating centres enabled staff at participating sites to focus on innovation and implementation of improvement interventions. The centres also functioned to coordinate networks of knowledge-sharing and exchange, linking sites with shared quality challenges together with the aim of accelerating the sharing of practices. They, therefore, played a critical role in marshalling the knowledge offered by local sites, sustaining the commitment and coordinating cooperation among participating organisations and clinicians.

We hook them up with somebody who was an outlier in the past who's not an outlier anymore, say, you might want to talk to so-and-so, because often the problem is not knowing the data, it's how do you apply best practices in your patient workflow. (Physician)

### High-quality measurement and comparative performance feedback

Participants emphasised that high-quality measurement and data were a prerequisite for successful collaborative quality improvement. Without credible data, participants reported that clinicians could neither reliably identify quality challenges, nor track the extent to which interventions (including those codified by positive deviance learning) had an impact. To secure legitimacy among clinicians, the collaboratives engaged deeply in processes to foster clinical ownership of data collection and analysis processes, underpinned by rigorous approaches to standardisation, risk adjustment, audit and quality assurance. Registry data then provided clinicians with a 'warrant for action' to change practice locally.

I have total confidence that I can go to my hospital or my people and say, this is real, we have a wound infection rate of 5.6, that's real, believe me, it's risk-adjusted, you can trust it, we got to work at it, there's a problem. (Surgeon)

Comparative performance feedback was key to making data meaningful and creating accountability for quality among clinical teams in participating hospitals. Alongside the provision of data on relative performance such as comparisons between providers and hospital-level rankings, forums for delivering feedback provided opportunities for participating clinicians to hold one another accountable for quality. These forums—including collaborative meetings, site visits and independent peer review of clinical practices—were also used to identify improvement opportunities and evaluate quality at the hospital and care provider level.

The leader of the coordinating centre will stand up and go, okay, Dr X, we see that you don't have a very good rate on your SSI [surgical site infections], so what's going on there? I mean, call them out. (Nurse)

### Careful use of motivational levers

The collaboratives employed a range of motivational levers to drive behaviour change among clinicians, including competitiveness, peer pressure and professional esteem. Healthy competition was seen as a key driver of improvement as clinicians were keen to be recognised by their peers as performing well.

There's nothing like a little bit of healthy competition. When a general surgeon sees that they're the lowest among their counterparts, that stimulates some serious ownership of that, and motivation to get better. (Nurse)

Besides their efforts to motivate participants, the collaboratives safeguarded friendly competition by introducing protections against anti-collaborative behaviours, thereby moderating the potential dark sides of excessive competitive tendencies. This was achieved, for example, by setting terms of reference that made clear that clinical registry data could not be used to gain commercial competitive advantage outside the collaborative community.

The other thing was that people wouldn't use the data for competitive advantage so that you would know your data, but you wouldn't be able to know who the other people were, and that nobody in the collaborative was allowed to publish or present the data or use the data at all from the collaborative to say that they did something better than some other group. (Surgeon)

### Mobilising professional leadership and building community

The collaboratives provided a formal framework through which clinicians could rally around a common purpose, giving expression to an underlying motivation to improve care for patients as a community of professionals. This required the creation and maintenance of an environment of trust, achieved by setting clear boundaries regarding data use, insulating improvement activities from the influence of insurance companies, building on pre-existing social networks, and protecting clinicians' autonomy.

They're the ones who see the data and the data does not go back to the insurance company. It's not used to decide where patients are directed. It's used to help surgeons to enable them to improve, and so that by really making sure that people began to trust the process and kind of following through on that and engaging in it they were making the decisions about what we were doing. (Surgeon)

The collaboratives cultivated high levels of trust between participating clinicians, giving them licence to experiment and to deviate from established practices in pursuit of ambitious improvement goals. Being part of a collaborative community provided protections necessary to take calculated risks to improve patient care.

[In reference to an innovative practice] Sites bought into that because they had trust to say, 'if everyone else in the group is doing this, I'm willing to go along as well. Even though I'm worried it may cause problems with bleeding, and this and that, I'm willing to give it a try because I trust the rest of this group enough.'" (Physician)

## DISCUSSION

Our study has identified five features of a high-performing state-wide collaboratives programme that may be implicated in its success: learning from positive deviance; high-quality coordination; high-quality measurement and comparative performance feedback; careful use of motivational levers; and mobilising professional leadership and building community. By providing insight into what good looks like, the findings are likely to be useful to policy-makers and healthcare systems in understanding how collaboration-based approaches can be optimised to maximise impact on quality of care, may help to address the persistent evidence gap about the features of collaboratives that drive performance, and could enable better replication and scaling of successful models.[38–40]

Our study underscores the importance of high-quality measurement in persuading clinicians of the veracity and actionability of quality problems. Broadly understood as the means by which performance in relation to quality of care can be reliably ascertained,[41 42] high-quality measurement has long been recognised as foundational to successful quality improvement efforts.[43–45] A well-established challenge for those using administrative data sources for improvement work[46–48] lies in ensuring that data are perceived by those being asked to act on it as valid, reliable and clinically relevant.[42] Doubts about data can lead to scepticism about the reality of problems and about the value of proposed solutions, or to loss of confidence in the improvement programme.[49–52] By implementing rigorous, clinically-led processes to risk-adjust and assure the quality of data used for improvement work, the Michigan collaboratives succeeded in avoiding many of the problems commonly associated with quality measurement[42 53] including the potentially weak risk adjustment often characteristic of administrative or insurance-based data sources,[48] problems with case capture and data completeness,[43 44 54] and variation in data collection systems.[41 42 55]

Accordingly, our findings indicate that those leading collaboratives should focus on optimising the credibility of data used for improvement. This might be achieved by ensuring data collection is aligned with the goals of participating professionals, and by bestowing responsibility for data quality on clinicians. Building on existing evidence of the advantages of securing professional leadership for data collection and the development of improvement interventions,[56] such an approach might better support collaborative improvement by strengthening clinical ownership.[54 57]

Consistent with previous research on the challenges of measurement for clinical teams, our study indicates that the task of securing clinical engagement for collaborative quality improvement is best achieved when the many mundane but highly consequential practical and technical challenges of data collection and analysis are taken care of in a competent way that relieves burden from participating clinicians.[43 54 56] By providing the knowledge and technical skills needed to collect, analyse and interpret clinical data for use in quality improvement,[58] collaboratives can address the persistent problem of shortages of local analytical capability,[43 59] support the identification of practices associated with high performance, and gain the confidence of clinicians. Analytics and improvement knowledge of this kind is increasingly recognised as an important form of expertise in its own right, and one that specialists other than clinicians may be best placed to provide.[60–62]

Equally important to successful collaboratives are conditions that enable a 'collaborative culture', including professional commitment to improvement and a spirit of collegial competitiveness, to flourish.[4 6 14 63 64] Previous research has drawn attention to the limitations of audit and performance feedback as an improvement strategy in healthcare,[65] including leveraging competition between hospitals and providers as stimulus for improvement.[66–69] A key lesson from our analysis is that making these levers work well-meant taking active steps to ensure that more malign forms of competitiveness were excluded. For example, safeguards such as a ban on the use of registry data to gain commercial advantage and on sharing unblinded performance data with insurance companies helped to preserve the collaboratives as communities where professional collegiality was the guiding principle. By identifying the role of protections against behaviours that undermine collaboration, our study highlights how collaboratives might successfully neutralise anti-competitive elements of programmes that can undermine improvement,[70 71] while activating the power of professional communities by insulating them from the negative effects of markets and bureaucracies.[72 73]

Finally, structures that create accountability for quality among clinicians appear to be critical.[4 63 74–76] In our case study, forums that brought participating clinicians together created 'obligations to explain and justify conduct',[77 78] establishing social relations that facilitated accountability for improving quality. This feature of the collaboratives was consistent with theory from social science that indicates that 'accountability requires a forum'[77]—a setting in which to *perform* accountability through social interaction—and may be especially

valuable in enhancing the effectiveness of performance feedback as an improvement strategy.

Our study has limitations. First, the use of qualitative methods and purposive sampling of participants, though appropriate to our goal of characterising the features of the collaboratives programme, necessitated a small sample of stakeholders, meaning that some selection bias may have occurred. Second, although our sample was intended to capture variability in perspectives among participants, practical considerations meant it was not possible to sample from all CQIs in the programme. Relatedly, surgical collaboratives—particularly the MSQC—and surgeon participants were over-represented in our interview sample, though the sample overall was broadly reflective of the target population (participants in the Michigan CQIs programme), where surgical CQIs accounted for the largest proportion of active collaboratives. Third, in common with all qualitative research, it is possible that researcher-related factors biased our analysis in some non-transparent way. Finally, our study did not include patient perspectives, which might have differed from those of participants; eliciting them may represent both a locus for improving the value of the CQIs programme and a focus of future research.

## CONCLUSIONS

Our study has identified 'what good looks like' for high-performing quality improvement collaboratives, including distinctive features of a state-wide programme. Our findings have implications for the design, structure and resourcing of quality improvement collaboratives both in the USA and internationally, and may inform efforts to replicate and scale successful models. Of particular importance is the need for rigorous measurement, securing professional leadership and engagement, cultivating a collaborative culture, creating accountability for quality and ensuring that collaborative infrastructures are adequately resourced to relieve participating sites of the burdens of improvement work. Just as important as technical and administrative support is the building of the right kind of community ethos: high-quality multi-organisational approaches to improving quality rely on norms and rules of engagement that enable both cooperation and healthy (non-financial) competition to flourish.

**Contributors** MD-W is the guarantor of the study. Concept and design: JGM, GLK, DAC and MD-W. Acquisition, analysis or interpretation of data: JGM, GPM, GLK and MD-W. Drafting of the manuscript: JGM, GPM and MD-W. Critical revision of the manuscript for important intellectual content: all authors. Statistical analysis: not applicable. Obtained funding: JGM and MD-W. Administrative, technical or material support: JGM, GLK, DAC, MJE and MD-W. Supervision: MD-W.

**Funding** This study was funded by James McGowan's NIHR Academic Clinical Fellowship (ACF-2016-14-011), by MD-W's Wellcome Trust Investigator award (WT097899) and by the Health Foundation's grant to the University of Cambridge for The Healthcare Improvement Studies (THIS) Institute. THIS institute is supported by the Health Foundation—an independent charity committed to bringing about better

health and healthcare for people in the UK. MD-W is a National Institute for Health Research (NIHR) senior investigator (NF-SI-0617-10026).

**Disclaimer** The funders had no role in the design and conduct of the study; collection, management, analysis and interpretation of the data; preparation, review, or approval of the manuscript; and decision to submit the manuscript for publication. The views expressed in this article are those of the authors and not necessarily those of the NHS, the NIHR, or the Department of Health and Social Care.

**Competing interests** The Michigan Quality Collaboratives are supported by Blue Cross and Blue Shield of Michigan. MJE receives salary support from Blue Cross and Blue Shield of Michigan as well as the NIH and the Michigan DHHS. DAC, GLK and MJE's institution (University of Michigan) receives a partial salary paid for by the Blue Cross Blue Shield of Michigan Value Partnerships for the Michigan Surgical Quality Collaborative. GLK, DAC and MJE are employed by the Michigan Surgical Quality Collaborative. MJE and JBD were interviewed as participants in the study.

**Patient and public involvement** Patients and/or the public were not involved in the design, or conduct, or reporting, or dissemination plans of this research.

**Patient consent for publication** Not applicable.

**Ethics approval** This study involves human participants but was exempted from review by the Health Sciences and Behavioral Sciences Institutional Review Board, University of Michigan (IRB-HSBS) (Study Ref: RG92221). All participants gave written informed consent to participate in the study.

**Provenance and peer review** Not commissioned; externally peer reviewed.

**Data availability statement** All data relevant to the study are included in the article or uploaded as online supplemental information. All data relevant to the study are included in the article or uploaded as online supplemental information. No additional data are available.

**ORCID iDs**
James G McGowan http://orcid.org/0000-0001-6963-4106
Graham P Martin http://orcid.org/0000-0003-1979-7577
Greta L Krapohl http://orcid.org/0000-0001-7409-1433
Darrell A Campbell http://orcid.org/0000-0003-0855-6145
Michael J Englesbe http://orcid.org/0000-0001-8691-9111
Justin B Dimick http://orcid.org/0000-0002-4796-6641
Mary Dixon-Woods http://orcid.org/0000-0002-5915-0041

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
