## [Reviewer comments · BMJ Open]

ARTICLE DETAILS

TITLE (PROVISIONAL)	What are the features of high-performing quality improvement collaboratives? A qualitative case study of a state-wide collaboratives programme
AUTHORS	McGowan, James; Martin, Graham; Krapohl, Greta; Campbell, Darrell; Englesbe, M; Dimick, Justin; Dixon-Woods, Mary

VERSION 1 – REVIEW

REVIEWER	Martin-Misener, Ruth Dalhousie University
REVIEW RETURNED	28-Aug-2023

GENERAL COMMENTS	This is a well-written article about an important topic. My comments are intended to strengthen the article. 1. A bit more background information about collaborative quality initiatives would be helpful for international readers. How widespread is their usage internationally and within the USA? When were they initiated and how have they evolved?2. The study was deemed exempt from ethics review yet written informed consent was obtained. Please explain why.3. Was this a single case study or multiple case study? My read of the article is that it is a single case study, but it would be helpful to clarify this and state it clearly.4. It sounds like you used purposive and maximum variation sampling for CQIs. How were the sites and individual participants recruited and by whom? For example, was there a letter /email of invitation?5. Clarify who conducted the interviews and if they were known to participants.6. Clarify who did the data analysis and if/how team members were involved in developing the framework for analysis and or coding the data.7. Please clarify what is the difference between and theme and a feature and the process that was used to evolve from theme to feature.8. Although nurses and managers were interviewed, it is difficult to hear their voice/perspective in the results, including quotes. It would be helpful if you could make their voices more evident. This could include identifying if a quote is from a nurse, physician or manager when transcripts are identified by number following a quote used in the paper.9. It seems like the article is mainly about physician performance and mostly surgical. If that is the case, it would be good to state that upfront. If the article is intended to be reflective of the performance of the team, consider including data and commentary to make that more obvious and or comment on any limitations that might exist with regard to availability of data that measures nursing and/or
---

	managerial performance.
--	-------------------------

REVIEWER	James, Brent Stanford University, Medicine
REVIEW RETURNED	01-Sep-2023

GENERAL COMMENTS	1. You repeatedly identify the Michigan Collaborative Quality Initiatives as a "high-performing regional collaboratives" program. However, you present no data or argument to justify that conclusion. Why should a reader believe the MCQIs falls into that category? 2. This paper really needs a careful and thorough edit to improve readability. 3. On page 5, lines 37-40, you note ethics oversight provided by a Univ of Michigan IRB. That is redundant with page 6, lines 44-49 (see #2 above). At page 5, line 48, you state that you 'purposively sampled clinicians and managers'. However, you never describe that method. What do you mean by 'purposively sampled'? At the same time, your Table 1 shows significant sample size differences across the various categories you list. This needs more explanation. For example, why does the Michigan Surgical Quality Collaborative dominate all other collaboratives, in terms of participant counts? 4. On page 6, line 19, you note that 'JM' (one of the authors) conducted participant observation. What does that mean? Is there any associated framework or structure? 5. On page 6, line 35-36, you stated that transcripts were read then 'coded and recoded multiple times'. How were they coded, and why were they recoded? As written, this leaves open the question of structural and analytic bias. It needs clarification. 6. While you cite the importance of 'high-quality measurement' in several places, your description of what that means and entails is insufficient to support that case. Can you add some a priori definitions, then outline your data and analytics structure within that framework (or use some other, more detailed, method). 7. You cite 'comparative performance data' and the 'relative performance such as provider and site-level rankings' as keys factors in a successful collaborative. As written, your presentation does not appear to appreciate the technical statistical literature associated with the limitations of ranking systems (see, for example, Andersson, Carling, and Mattson. Random ranking of hospitals is unsound. Chance 1998; 11(3):34-37,39). You could greatly strengthen this section by showing how you appropriately addressed such issues. Your description of 'high-quality measurement' appears in the Discussion (lines 15-43, page 17). It needs to be extended (see above), and probably should be placed much earlier in your article, before you cite and rely on it.
--

VERSION 1 – AUTHOR RESPONSE

Reviewer 1 Comments		
1. A bit more background information about collaborative quality initiatives would be helpful for international readers. How widespread is their usage internationally and within the USA? When were they initiated and how have they evolved?	Thank you for this suggestion to strengthen the background to our paper. We have amended and extended the Introduction section to address this feedback.	5-6
2. The study was deemed exempt from ethics review yet written informed consent was obtained. Please explain why.	The study was deemed exempt from review by an Institutional Review Board. This is separate from our duty to ensure participants were given full information about the nature of the study, and consented to participate in it.	N/a
3. Was this a single case study or multiple case study? My read of the article is that it is a single case study, but it would be helpful to clarify this and state it clearly.	We conducted a single case study of a regional (state-wide) collaboratives programme. The programme comprised multiple quality improvement collaboratives covering a range of surgical and medical specialties. We have made changes to the Abstract, Introduction and Methods sections to clarify this point.	2-3, 5-9
4. It sounds like you used purposive and maximum variation sampling for CQIs. How were the sites and individual participants recruited and by whom? For example, was there a letter /email of invitation?	Thank you – we have expanded the Methods section to include further detail on how sampling and recruitment were performed in this study, including the criteria by which we sampled, and the way in which potential participants were approached.	7-9
5. Clarify who conducted the interviews and if they were known to participants.	All interviews were conducted by one interviewer (JM) who was not known to the participants – this has been clarified in the Methods section of the manuscript.	8
6. Clarify who did the data analysis and if/how team members were involved in developing the framework for analysis and or coding the data.	Thank you – we have added a clarification in the Methods section regarding responsibility for data analysis and development of the thematic framework.	9
7. Please clarify what is the difference between and theme and a feature and the process that was used to evolve from theme to feature.	Thank you for highlighting the need for clarity on this point. The ‘themes’ identified in Table 2 may be more accurately described as data categories – they are conceptually related to each other and to the overarching features under which they appear in table. Given we refer to our features as a ‘thematic framework’ in the manuscript, we have sought to clarify this distinction by re-labelling the ‘theme’ column as ‘data categories’- we hope this clarifies the distinction between these concepts.	Table 2, Page 11

	Regarding the analytical process that supported development of the thematic framework, qualitative analysis of the dataset was based on the constant comparison method described by Charmaz 2006 (cited in the manuscript). In employing this method, our coding process involved making series of analytic comparisons between data, codes and categories of codes in order to identify distinct analytic concepts in the data and relationships between them. This process was inductive and required iterative coding and re-coding of the data multiple times to develop our analysis, and ultimately to synthesise the conceptual framework that is presented in our results section (and in Table 2).	
8. Although nurses and managers were interviewed, it is difficult to hear their voice/perspective in the results, including quotes. It would be helpful if you could make their voices more evident. This could include identifying if a quote is from a nurse, physician or manager when transcripts are identified by number following a quote used in the paper.	Thank you for this suggestion to improve the balance and transparency of our reporting. We have re-labelled quotes as coming from a surgeon, physician, nurse or manager. Additionally, we have added new quotations from nurse participants to improve the balance of reporting, and amended table 1 to highlight that both surgeons and physicians were interviewed. The breakdown of quotes included in the paper (including table 2) is as follows: Surgeons: 10 quotes Physicians: 2 quotes Nurses: 7 quotes Managers: 6 quotes	Throughout manuscript, including Tables 1 and 2
9. It seems like the article is mainly about physician performance and mostly surgical. If that is the case, it would be good to state that upfront. If the article is intended to be reflective of the performance of the team, consider including data and commentary to make that more obvious and or comment on any limitations that might exist with regard to availability of data that measures nursing and/or managerial performance.	Thank you for this feedback. The study did not set out to focus either on physician performance to the exclusion of other team members, or on surgical collaboratives at the expense of other specialities. Rather, we sought to identify and characterise features of a high-performing group of collaboratives that may be implicated in their ability to achieve improvements in quality of care. As both reviewers have noted, both surgeons and surgical collaboratives (particularly the Michigan Surgical Quality Collaborative) are over-represented in our interview sample. This is broadly reflective of the target population (participants in all CQIs in the Michigan programme), where surgical CQIs accounted for the largest proportion of active collaboratives. As the largest CQI with the most members across the state, the fact that many interview participants were members of the Michigan Surgical Quality Collaborative is also not surprising. We have added text both to the Methods section and to the limitations paragraph of the Discussion to acknowledge and contextualise this point.	16

	Although our focus was on understanding drivers of performance of the collaboratives as a whole (rather than of individual physicians), we found performance comparison among both physicians and hospitals (implying clinical teams) to be a key method employed by the CQIs to drive quality improvement – this is reflected in our results. We have amended the text on page 16 slightly to emphasise that hospital-level site rankings imply performance assessment for whole clinical teams, not only physicians. Finally, synthesis of our thematic framework was based on analysis of the whole dataset of interview transcripts. Given that most (60%) of our interviewees were nurses or managers, we believe our framework is likely to reflect their perspectives to a proportionate extent.	
Reviewer 2 Comments		
You repeatedly identify the Michigan Collaborative Quality Initiatives as a "high-performing regional collaboratives" program. However, you present no data or argument to justify that conclusion. Why should a reader believe the MCQIs falls into that category?	Thank you for this suggestion to strengthen the context for our study- we have amended and extended the Introduction section to address this feedback.	5-6
This paper really needs a careful and thorough edit to improve readability.	Thank you- we have reviewed the manuscript carefully and corrected errors where we could identify them, including the instance of duplication cited below.	Throughout manuscript
On page 5, lines 37-40, you note ethics oversight provided by a Univ of Michigan IRB. That is redundant with page 6, lines 44-49 (see #2 above).	This duplication error has been addressed in the manuscript.	7-9
At page 5, line 48, you state that you 'purposively sampled clinicians and managers'. However, you never describe that method. What do you mean by 'purposively sampled'? At the same time, your Table 1 shows significant sample size differences across the various categories you list. This needs more explanation. For example, why does the Michigan Surgical Quality Collaborative dominate all other collaboratives, in	Thank you for raising these important points regarding sampling. We employed a purposive sampling strategy aiming to achieve a balance of participants across a range of characteristics listed in the Methods section. We have added further detail regarding how participant recruitment was conducted (on page 7). As the reviewer notes, surgical collaboratives – particularly the Michigan Surgical Quality Collaborative (MSQC) – are over-represented in the interview sample. This broadly reflected the target population (participants in all CQIs in the Michigan programme), where surgical CQIs accounted for the largest proportion of active collaboratives. As the largest CQI with the most participants across the state, the finding	7, 21

terms of participant counts?	that many interview participants were members of MSQC would be expected. We have added text both to the Methods section and to the limitations paragraph of the Discussion to acknowledge and contextualise this point.	
On page 6, line 19, you note that 'JM' (one of the authors) conducted participant observation. What does that mean? Is there any associated framework or structure?	We employed participant observation as a method of collecting qualitative data as part of our case study. We have added more detail in the Methods section regarding the method and the setting in which observation was conducted.	8
On page 6, line 35-36, you stated that transcripts were read then 'coded and recoded multiple times'. How were they coded, and why were they recoded? As written, this leaves open the question of structural and analytic bias. It needs clarification.	Qualitative analysis of our dataset was based on the constant comparison method described by Charmaz 2006 (cited in the manuscript). In employing this method, our coding process involved making a series of analytic comparisons between data, codes and categories of codes in order to identify distinct analytic concepts in the data and relationships between them. This process was inductive and required iterative coding and re-coding of the data multiple times to develop our analysis, and ultimately to synthesise the conceptual framework that is presented in our results section. We have added further detail in the Methods section regarding our analytical approach.	8-9
While you cite the importance of 'high-quality measurement' in several places, your description of what that means and entails is insufficient to support that case. Can you add some a priori definitions, then outline your data and analytics structure within that framework (or use some other, more detailed, method).	Thank for your suggestions to improve our discussion of high-quality measurement in the context of our study. We have added additional text and citations in the Discussion section to address this feedback.	19
You cite 'comparative performance data' and the 'relative performance such as provider and site-level rankings' as keys factors in a successful collaborative. As written, your presentation does not appear to appreciate the technical statistical literature associated with the limitations of ranking systems (see, for example, Andersson, Carling, and Mattson. Random ranking of hospitals is unsound. Chance 1998; 11(3):34-	Thank you for this helpful suggestion. Though one focus of our discussion is on how a 'collaborative culture' rather than competition per se supported improvement, we appreciate that acknowledging prior literature on limitations of comparative performance feedback may strengthen the context in which our findings are presented. We have added text and citations to the Discussion section to address this feedback. Additionally, we acknowledge the limitations of ranking as a specific mode of presenting and interpreting performance data – for example, the way in which it can over-emphasise small differences in performance between institutions	20

37,39). You could greatly strengthen this section by showing how you appropriately addressed such issues.	which may reflect random variation. As we discuss in the paper, performance feedback in the CQIs was based on registry data which was perceived by clinicians to be rigorously risk-adjusted, thus dealing (in their view) with a major concern regarding the validity of data on which rankings were based. Presentation of comparative performance data was mostly blinded/anonymised by the coordinating centres, though some CQIs opted to unblind their data to clinicians. Naturally, our findings reflect only what participants believed to be important drivers of improvement in their particular context of the Michigan programme. Our findings are not necessarily generalisable, and may be in tension with previous research conducted in other settings. In our study, the perceived effectiveness of comparative performance feedback (including rankings) as a driver of improvement may be an example of such a tension.	
Your description of 'high-quality measurement' appears in the Discussion (lines 15-43, page 17). It needs to be extended (see above), and probably should be placed much earlier in your article, before you cite and rely on it.	Please see our response above regarding improvements made to our discussion of high-quality measurement. As 1 of 5 features in our thematic framework, we believe dedicating two paragraphs in the Discussion section to discussing issues of measurement in improvement is proportionate to its importance in our analysis. To ensure balance, we do not wish to place undue emphasis on measurement at the expense of other features of our framework.	19

VERSION 2 – REVIEW

REVIEWER	Martin-Misener, Ruth Dalhousie University
REVIEW RETURNED	18-Nov-2023
GENERAL COMMENTS	Thank you for your clear and illuminating revisions.